# Synthetic Microbial Communities Enhance Pepper Growth and Root Morphology by Regulating Rhizosphere Microbial Communities

**DOI:** 10.3390/microorganisms13010148

**Published:** 2025-01-13

**Authors:** Tian You, Qiumei Liu, Meng Chen, Siyu Tang, Lijun Ou, Dejun Li

**Affiliations:** 1College of Horticulture, Hunan Agricultural University, Changsha 410125, China; sidadiyou@outlook.com (T.Y.); chenmeng@stu.hunau.edu.cn (M.C.); 2Hunan Provincial Key Laboratory of Agroecological Engineering, Key Laboratory of Agro-Ecological Processes in Subtropical Region, Institute of Subtropical Agriculture, Chinese Academy of Sciences, Changsha 410125, China; liuqiumei@isa.ac.cn (Q.L.); siyutang@isa.ac.cn (S.T.); 3Guangxi Key Laboratory of Karst Ecological Processes and Services, Huanjiang Observation and Research Station for Karst Ecosystems, Chinese Academy of Sciences, Huanjiang 547100, China

**Keywords:** synthetic microbial community, pepper growth, microbial community, key taxa

## Abstract

Synthetic microbial community (SynCom) application is efficient in promoting crop yield and soil health. However, few studies have been conducted to enhance pepper growth via modulating rhizosphere microbial communities by SynCom application. This study aimed to investigate how SynCom inoculation at the seedling stage impacts pepper growth by modulating the rhizosphere microbiome using high-throughput sequencing technology. SynCom inoculation significantly increased shoot height, stem diameter, fresh weight, dry weight, chlorophyll content, leaf number, root vigor, root tips, total root length, and root-specific surface area of pepper by 20.9%, 36.33%, 68.84%, 64.34%, 29.65%, 27.78%, 117.42%, 35.4%, 21.52%, and 39.76%, respectively, relative to the control. The Chao index of the rhizosphere microbial community and Bray–Curtis dissimilarity of the fungal community significantly increased, while Bray–Curtis dissimilarity of the bacterial community significantly decreased by SynCom inoculation. The abundances of key taxa such as *Scedosporium*, *Sordariomycetes*, *Pseudarthrobacter*, *norankSBR1031*, and *norankA4b* significantly increased with SynCom inoculation, and positively correlated with indices of pepper growth. Our findings suggest that SynCom inoculation can effectively enhance pepper growth and regulate root morphology by regulating rhizosphere microbial communities and increasing key taxa abundance like *Sordariomycetes* and *Pseudarthrobacter*, thereby benefiting nutrient acquisition, resistance improvement, and pathogen resistance of crops to ensure sustainability.

## 1. Introduction

Pepper is the most widely cultivated vegetable in China, accounting for a cultivating area of 827,000 hectares and approximately 46% of the world’s total pepper production [1,2]. In recent years, the expansion of intensive agriculture has resulted in the unreasonable use of chemical fertilizers and pesticides, along with prolonged continuous cropping practices. These factors have caused decreased pepper yield per unit area, severe disease outbreaks, and deteriorated soil quality, threatening the sustainable development of the pepper industry [3,4,5]. Beneficial microbial inoculants can promote plant growth through various mechanisms, such as nutrient enhancement, plant hormone production, iron carrier secretion, microbial community assembly, activation of native microbes, synthesis of antimicrobial compounds, and induction of plant resistance [6,7]. However, due to the complex field conditions, the application of SynCom often results in the limited functionality, poor colonization and weak survival rate of inoculants [8]. Therefore, there is an urgent need to explore more effective strategies to increase the efficiency of inoculants to meet the demands of sustainable development of vegetables.

Beneficial microbial microbes of soil play a crucial role in promoting plant growth through various mechanisms, such as nutrient enhancement, plant hormone production, iron carrier secretion, microbial community assembly, activation of native microbes, synthesis of antimicrobial compounds, and induction of plant resistance [6]. Previous studies have highlighted that beneficial microbe inoculation at the seedling stage is an effective method to promote plant growth, reduce disease and increase plant resistance [9,10]. For example, *Bacillus amyloliquefaciens* and *Bacillus subtilis* were found to promote seed germination, cell development and seedling growth by altering microbial community composition and function [11,12]. *Arthrobacter nicotionovorans* and *Pseudomonas mucilaginosa* were found to stimulate cucumber, pepper and ginseng growth through processes such as IAA biosynthesis, phosphorus solubilization, potassium release, and nitrogen fixation, significantly impacting the composition and structure of rhizosphere microbial communities [13]. *Trichoderma longibrachiatum* and *Trichoderma viride* exhibit hyperparasitism on plant roots by producing various enzymes and signaling substances, promoting cucumber growth, pathogen cell lysis, and host plant systemic resistance [14,15,16]. The interaction between *Trichoderma harzianum* and plants can produce signaling substances such as peroxidase, organic acids, amino acids, chitinase, β-1,3-glucanase, volatile organic compounds (VOCs), shikimic acid terminal secondary metabolites, and ATP-binding osmotic transporters, which regulate plant architecture and reduce the accumulation of host cell toxins [17,18]. The metabolic characteristics of the interaction between inoculants and plant roots are essential for promoting the assembly of rhizosphere microbial communities and ensuring their growth-promoting functions [19]. For instance, *P. fluorescens* can secrete volatile organic compounds, including antitoxin terpenes and other compounds, to promote plant growth and combat pathogens [20,21]. Similarly, *Trichoderma* biosynthesis contributes to inducing complex signal network responses by producing substances like salicylic acid, jasmonic acid, ethylene, nitric oxide, avirulence effectors, and small secreted cysteine-rich proteins, which promote plant development and the expression of defense-related genes [22]. However, the effectiveness of inoculants in promoting plant growth may be unstable due to their weak survival and colonization capacities in soil [23].

SynCom application benefits soil health and crop productivity via pathogen suppression, recruitment of beneficial native microbes, modification of rhizosphere microbiota composition, and the enhancement of plant stress resistance, thus decreasing the use of chemical fertilizers and pesticides [24,25]. Compared to single functional strains, the inoculation of SynCom into seedling substrates and soil has been found to be more efficient in promoting crop growth by broadening the types of metabolites and improving the synergistic effects of microbial community assembly [26,27]. Li et al. [28] developed a SynCom that efficiently stimulated root biomass and prevented Astragalus root rot disease by regulating microbial communities with high-abundance bacteria responsible for enhancing plant growth and root morphology, while low-abundance bacteria inducing systemic resistance. SynComs have demonstrated significant growth-promoting effects in a variety of vegetable crops, including tomatoes, cucumbers, and watermelons, by enhancing soil nitrogen content, secreting plant growth regulators, and improving nutrient uptake efficiency, soil microbial community structure, and biological activity [29]. Synthetic microbial communities exhibit complementary effects in utilizing nutrient resources, synergistically readjusting the structure of rhizosphere microbial communities, restoring community diversity, and enhancing the prevalence of beneficial indigenous microbial strains, thus promoting plant growth and development [30,31]. A meta-analysis revealed that the application of synthetic microbial communities notably increased plant growth compared to single-strain inoculation [32]. The combination of *Trichoderma harzianum*, *Bacillus subtilis*, and *Rhizobium leguminosarum* biovar viciae can cooperatively regulate root development and increase the yield of fava beans [33]. The combination of *Trichoderma viride* and *Trichoderma harzianum* significantly enhances leaf area index, leaf number, shoot length, and root length of black pepper [34]. However, the effect of SynCom inoculation on pepper growth remains largely unknown to date [35].

In the current study, the inoculation of a SynCom composed of *Bacillus subtilis*, *Trichoderma harzianum*, *Trichoderma asperellum* and *Aspergillus* sp. into pepper seedling substrates was conducted to assess its effectiveness in promoting pepper growth. The primary objectives were to address the following questions: (1) How efficient is the inoculation of the SynCom for promoting pepper growth, and (2) what are the characteristics of the rhizosphere microbial community responsible for enhancing pepper growth?

## 2. Materials and Methods

### 2.1. Preparation of SynCom

The SynCom consist of *Bacillus subtilis*, *Trichoderma harzianum*, *Trichoderma asperellum* and *Aspergillus* spp. from the Institute of Subtropical Agriculture, Chinese Academy of Sciences. These strains were assigned the storage numbers GDMCC 64986, GDMCC 64988, GDMCC 65053, and GDMCC 65054, respectively, and were deposited on 7 August 2024, at Guangdong Microbial Culture Collection Center (GDMCC). The GDMCC is situated on the 5th floor of Building 59, No. 100 Xianlie Middle Road, Guangzhou. Each strain was mixed in equal volumes with sterilized 30% glycerol and stored at −80 °C to prevent strain degradation.

2.0 μL of fungal spore solution stored at −80 °C was inoculated onto the central area of Potato Dextrose Agar (PDA) plates for cultivation at 28 °C until conidiospore germination. Subsequently, 10 mL of sterile water was added to the PDA plate containing fungal spores. The fungal spores were obtained by filtering the solution through four layers of sterile gauze to prepare a spore suspension. The spore concentration was determined using a hemocytometer. A fungal solution with a spore concentration was then inoculated into a 250 mL conical flask containing 100 mL of sterilized liquid PDA medium. The flask was cultured at 28 °C and 170 rpm for 5 days to produce the fungal fermentation liquid. Similarly, 2.0 μL of bacterial solution stored at −80 °C was inoculated onto the central area of Luria Broth (LB) agar plates for cultivation at 37 °C for 8 h until single colonies grew. Single bacterial colonies were selected and inoculated into 250 mL conical flasks containing 100 mL of sterilized liquid LB medium. The flasks were then cultured at 37 °C and 170 rpm for 24 h to obtain the bacterial fermentation liquid. The fermentation liquids of the four strains were adjusted to a concentration of 1 × 10^8^ CFU/mL, and then mixed in equal proportions to create the SynCom.

### 2.2. Experiment of Pepper Growth

Pepper seeds were surface-sterilized with 75% (*v*/*v*) ethanol for 1 min, followed by treatment with 10% (*v*/*v*) sodium hypochlorite for 20 min. Subsequently, the seeds were rinsed with distilled water five times. The pepper seeds were placed in sterile Petri dishes containing 0.5× Murashige Skoog (MS) medium and incubated in the dark at 30 °C for 36 h for germination. After germination, the seeds were cultured in a humidity-controlled plant growth chamber with 0.5× MS solution for 7 days. The culture conditions were maintained at 28 °C, at 80% humidity, with a 16 h light and 8 h dark cycle. Two treatments were designed: control and SynCom inoculation. The seedling substrates were obtained from the local agricultural inputs market and filled into a plug seedling. One hundred germinated seedlings with similar root and shoot sizes were placed in each tray containing 100 g of substrate for each treatment. Each treatment consisted of three replicates of seedling trays. They were cultivated for 5 days in an artificial climate chamber with the following conditions: 16 h of light at 28 °C and 8 h of dark at 28 °C, ensuring a photosynthetic photon flux density of 300 mmol m^2^ s^−1^ and a constant relative humidity of 80%. After 5 days of cultivation, the newly prepared SynCom was inoculated into soil at a concentration of 1 × 10^7^ CFU/g. Following 45 days of cultivation, six plants were randomly selected from each treatment to measure pepper growth-related indicators, conduct root scanning, determine root vitality, and collect rhizosphere soil samples for DNA extraction for subsequent high-throughput sequencing analysis.

Soil samples from the rhizosphere were obtained by vigorously shaking the plant roots to separate the loosely attached soil, collecting approximately 1 mm of soil remaining adhered to the roots. The roots with soil still attached were placed in a sterile flask with 50 mL of sterile phosphate-buffered saline (PBS) solution. The roots were then vigorously stirred with sterile forceps to remove all soil from the root surfaces. The soil suspension was centrifuged at 8000 rpm for 20 min, and the resulting pellet soil was defined as the rhizosphere soil.

### 2.3. Determination of Pepper Growth and Root Morphology

Pepper shoot height is defined as the length from the base of the pepper stem to the growth point. Pepper stem diameter was determined using a vernier caliper 1 cm above the base of the stem. The number of leaves were calculated by counting all leaves whose length exceeded half of the normal mature leaf length. To determine fresh weight, the plants were washed with sterile water, dried with absorbent paper, and weighed using an electronic balance. Dry weight was measured by weighing the plants after being placed in an oven at 105 °C for 15 min to halt enzyme activity, followed by drying at 60 °C to a constant weight. Chlorophyll content was assessed using a handheld chlorophyll meter (KONICA MINOLTA, Tokyo, Japan).

The root vigor was assayed using the Triphenyltetrazolium chloride (TTC) method as described by Clemensson-Lindell et al. [36]. Initially, 0.5 g of plant lateral roots were carefully washed with water and dried with filter paper. Subsequently, the roots were fully immersed in a TTC buffer and incubated at 37 °C in darkness for 2.5 h. The reaction was halted by adding 2 mL of stop solution. For each treatment, a corresponding blank control was prepared by immersing the roots in 2 mL of TTC stop solution to deactivate the root samples, followed by the addition of 10 mL of TTC buffer solution. The roots from the reaction solution were extracted, dried with filter paper, and then homogenized in a mortar with 3–4 mL of ethyl acetate solution. The extract was transferred to a centrifuge tube, and the remaining extracts were washed three times with 1 mL of ethyl acetate to reach a total volume of 10 mL. The absorbance was measured at a wavelength of 485 nm. A standard substance consisting of 2 mg of Na_2_S_2_O_4_ dissolved in 10 mL of ethyl acetate was used for calibration. Root dehydrogenase activity (mg/g/h) was calculated as follows: TTC content (μg)/[1000 × root weight (g) × time (h)].

To analyze the root morphology of pepper, the roots were cut from the stem base. The roots were washed with deionized water and placed in a transparent culture dish. A suitable amount of deionized water was injected for Plant Magnetic Resonance Imaging (MRI) using a flatbed scanner (SCAN-GXY-A, Beijing, China). The root morphology parameters, including total root length, total root tip number, and total surface area, were analyzed using the Detta-TSCAN root analysis system (Delta-T Device Ltd., Cambridge, UK).

### 2.4. High-Throughput Sequencing and Bioinformatics Analysis

Genomic DNA was extracted from 0.25 g of rhizosphere soil samples using the DNeasy PowerSoil Pro kit (QIAGEN, Valencia, CA, USA) following the manufacturer’s protocol. Additionally, genomic DNA extraction was carried out from each pepper root sample using the Plant DNA Maxi Kit (Omega Bio-Tek, Norcross, GA, USA) as per the manufacturer’s instructions. The concentration and purity of the extracted DNA were assessed to meet the requirements of quantitative real-time PCR amplification (qPCR) and Illumina sequencing using the NanoDrop ND2000 (Thermo Scientific, Waltham, MA, USA). For ITS amplicon sequencing, primers ITS1F (CTTGGTCATTTAGAGGAAGTAA) and ITS2R (GCTGCGTTCTTCATCGATGC) were utilized. 16S amplicon sequencing was conducted with primers 338F (ACTCCTACGGGAGGCAGCAG) and 806R (GGACTACHVGGGTWTCTAAT). These sequencing processes were performed by Shanghai Majorbio Bio-pharm Technology Co., Ltd., (Shanghai, China).

The raw data underwent preliminary analysis using UPARSE, Qiime2, and the RDP online analysis platform. The relative abundance of various microbial taxa in rhizosphere soil DNA samples was determined at a specific taxonomic level as a percentage of the total sequences obtained from each sample. Microbial community-richness index (Chao or Sobs), diversity indices (Shannon), and phylogenetic diversity indices (Faith’s PD) were calculated to compare the alpha diversity of microbial communities between the two treatments. Principal coordinate analysis (PCoA) was employed to compare the beta diversity of rhizosphere microbial communities between the two treatments based on Bray–Curtis dissimilarity.

### 2.5. Statistical Analysis

Student’s *t*-test was carried out to assess whether a significant difference existed at the *p* < 0.05 level between the two treatments, and Student’s *t* test (two-sided) was used to examine the difference in the variables between the two treatments. One-way analysis of variance (ANOVA) with the least significant difference (LSD) test was used to determine the difference in multiple comparisons. The outcomes were visualized utilizing the gplot package in R4.3.1. The error bars in figures represents standard error. The data displayed in the graphs were represented as mean values ± standard error. To investigate the connection between microbial community composition and diversity and the root development and growth of pepper seedlings, correlation analysis was conducted using the corrplot package in R software (version 4.0.5). The relative importance of selected key taxa in the growth and root development of pepper seedlings was analyzed utilizing the rfPermute package and the random forest package in R4.3.1.

## 3. Results

### 3.1. Effects of SynCom on Pepper Growth

SynCom inoculation significantly enhanced the growth of pepper seedlings (Figure 1a). Shoot height, stem diameter, fresh weight, dry weight, chlorophyll content, and number of leaves were significantly increased under SynCom inoculation by 20.9% ± 0.018, 36.33% ± 0.053, 68.84% ± 0.035, 64.34% ± 0.014, 29.65% ± 0.011, 27.78% ± 0.56, respectively, compared to the control after 45 days of cultivation (Figure 1b–g, *p* < 0.05). Furthermore, the inoculation of the SynCom notably improved root growth and development (Figure 2a,b). Specifically, the root vigor, root tips, total root length, and root surface area were significantly increased by 117.42% ± 0.092, 35.4% ± 0.052, 21.52% ± 0.028, and 39.76% ± 0.027, respectively, relative to the control after 45 days of cultivation (Figure 2c–f, *p* < 0.05). Conclusively, the inoculation of SynCom had a profound positive influence on pepper growth, activity of root vigor, and root morphology, ultimately enhancing pepper productivity and stress resistance.

### 3.2. Effects of SynCom Inoculation on Rhizosphere Microbial Community

SynCom inoculation significantly increased the Chao index of the rhizosphere microbial community (Figure 3a,c, *p* < 0.05). However, no significant differences were observed in the Shannon index and Simpson index of the rhizosphere fungal community between the two treatments (Figure 3b,d,f). The beta diversity of the rhizosphere microbial community exhibited significant changes under SynCom inoculation, including a notable increase in the Bray–Curtis dissimilarity of the fungal community and a decrease in the Bray–Curtis dissimilarity of the bacterial community (Figure 3g–j, *p* < 0.05). Regarding the bacterial community composition, the genera *Flavobacterium* and *Pseudarthorobacter* were most abundant for both treatments. SynCom inoculation significantly increased the composition of the rhizosphere community, but did not significantly altered microbial richness and evenness. In terms of the fungal community composition, the genera *Mycothermus* and *Sordariomycetes* were the most abundant for both treatments.

SynCom inoculation increased the abundances of *Sordariomycetes*, *Zopfiella*, *Arachniotus*, and *Roellomycota*, and decreased those of *Mycothermus*, *Byssochlamys*, *Scedosporium*, *Lobulomycetes* relative to the control (Figure 4a). For the bacterial community composition, SynCom inoculation increased the abundances of *Pseudarthorobacter*, *norank SBR1031*, *norank A4b*, and *Bacillus*, and decreased those of *Flavobacterium*, *Pseudomonas*, *Comamonadaceae*, *Massilia*, *Acidovorax*, and *Rhodococcus* (Figure 4b). Notably, the relative abundance of fungal *Sordariomycetes* increased significantly, whereas that of *Byssochlamys* and *Scedosporium* decreased significantly under SynCom inoculation (Figure 4c). Similarly, the relative abundance of bacterial *Pseudarthorobacter* increased significantly, while that of *Acidovorax* or *Rhodococcus* decreased significantly under SynCom inoculation (Figure 4d). Correlation analysis was conducted to explore the relationships among the top 10 genera of the bacterial and fungal community, followed by a Mantel test to investigate the potential effects of these genera on pepper growth and root morphology. The abundance of *Sordariomycetes*, *Pseudarthrobacter*, *norank SBR1031*, and *norank A4b* were significantly correlated with pepper growth and root morphology (Figure 4e). Therefore, the higher Chao index and Bray–Curtis dissimilarity of fungal diversity, along with the increased abundance of *Sordariomycetes*, *Pseudarthrobacter*, *norank SBR1031*, and *norank A4b* under SynCom inoculation, may play a crucial role in pepper growth and root morphology.

### 3.3. The Importance of Rhizosphere Microbial Community for Pepper Growth

A total of 1073 fungal and 7039 bacterial operational taxonomic units (OTUs) were identified for the control and SynCom inoculation treatments, respectively. The abundance of OTU451 classified as *Scedosporium*, OTU346, OTU439, OTU437, OTU381, OTU447, and OTU182 classified as *Sordariomycetes*, OTU2309 classified as *Pseudarthrobacter*, OTU3060, OTU2704 classified as *norank SBR1031*, and OTU3638, OTU3534, and OTU2118 classified as *norank A4b* were significantly increased, while OTU327 classified as *Rhodococcus* was significantly decreased by SynCom inoculation (Figure 5a,b).

Random forest analysis revealed that OTU451, OTU182, OTU346, OTU2309, and OTU3638 were identified as the most important contributors, and OTU3060, OTU2704, and OTU2118 also played important roles in shoot height (Figure 5c). Similarly, the abundance of OTU2309, OTU451, OTU2118, and OTU3060 were highlighted as key contributors, while OTU2704, OTU346, OTU182, and OTU3638 were significant in determining the dry weight of pepper (Figure 5d). Furthermore, OTU2118 and OTU182 emerged as prominent factors affecting chlorophyll content, whereas OTU346, OTU2704, OTU3638, OTU2309, OTU451, and OTU3060 played important roles as well (Figure 5e). The abundance of OTU3638, OTU2704, and OTU2118 significantly impacted the activity of root vigor (Figure 5f). Therefore, SynCom inoculation led to a substantial increase in the abundance of key species, including *Scedosporium* (OTU451), *Sordariomycetes* (OTU346, OTU439, OTU437, OTU381, OTU447, and OTU182), *Pseudarthrobacter* (OTU2309), *norank SBR1031* (OTU3060, OTU2704), and *norank A4b* (OTU3638, OTU3534, OTU2118), which played crucial roles in promoting pepper growth and root morphology.

## 4. Discussion

### 4.1. SynCom Inoculation for Plant Growth Promotion

In the current study, the inoculation of SynCom consisting of *Bacillus subtilis*, *Trichoderma harzianum*, *Trichoderma asperellum*, and *Aspergillus* sp. significantly increased shoot height, stem diameter, fresh weight, dry weight, chlorophyll content, leaf number, root vigor, root tips, total root length, and root-specific surface area of peppers, implying that the inoculation of these beneficial microbes plays a crucial role in promoting pepper growth and regulating root morphology. It has been suggested that *Bacillus subtilis*, *Trichoderma harzianum*, *Trichoderma asperellum*, and *Aspergillus* sp., have the capability to enhance plant growth, eliminate pathogens, and increase plant resistance when used as individual inoculants. For instance, *Bacillus subtilis*, a well-studied rhizosphere bacterium, has been shown to boost plant growth through mechanisms such as improving nutrient availability, modulating plant hormone levels, producing antibacterial agents, and inducing systemic resistance [37]. Similarly, the inoculation of *Bacillus subtilis* FJ3, has been proven to enhance growth in chickpeas by 24.01% [38]. Furthermore, *Trichoderma harzianum* inoculation has shown promising efficiency in significantly increasing tobacco stem length, root length, dry biomass, root surface area, and root tips, as well as elevating phenolic compound content and antioxidant activity in leaves [39]. The enhancement of pepper growth due to the addition of *Trichoderma harzianum* may be attributed to its ability to reduce the accumulation of host cell toxins, promote pathogen lysis, enhance plant systemic resistance, activate soil nutrients, modulate microbial communities and interactions, and stimulate plant growth and development through the production of various enzymes and signaling substances such as organic acids, proteases, plant hormones, VOCs, and amino acids during plant interactions that promote plant growth, development, and stress tolerance [40,41]. Overall, the increased pepper biomass, root vigor and root tips observed following the inoculation of beneficial microbes in the present study suggest that microbial inoculation may have effectively stimulated the synthesis of phytohormones, solubilization of minerals, production of siderophorus and induction of antioxidant enzymes, ultimately leading to an increase in nutrient uptake for plant growth promotion. The application of synthetic microbial communities is a crucial method to improve crop yield by enhancing the activity of microbial inoculants, regulating the structure of rhizosphere microbial communities, restoring community diversity, enriching beneficial native microorganisms, and broadening the types and functions of metabolites through complementary and synergistic effects of nutrient resource utilization among strains [42,43]. The variations in Bray–Curtis dissimilarity trends between fungal and bacterial communities arise from their unique ecological niches and functions. Bacteria, with their quick responses to environmental changes due to shorter generation times, play diverse roles as decomposers and pathogens, while fungi, particularly in mycorrhizal associations, have longer lifespans and stable communities, sensitive to specific environmental conditions. These contrasting dynamics underscore the complexity of microbial interactions and responses to environmental factors, highlighting the need for precise ecosystem management strategies to uphold ecosystem functioning. The significantly increased pepper growth, activities of root vigor, and regulation of root morphology following the addition of SynCom imply that SynCom might have changed rhizosphere microbial community composition, diversity and function, which subsequently promoted plant growth and development [44,45,46]. Our findings underscore the considerable potential of synthetic microbial communities as a novel strategy for ensuring plant growth in key agricultural crops.

### 4.2. SynCom Regulating Microbial Community and Key Taxa for Plant Growth

The present study showed that the relationship between pepper growth and root morphology was predominantly influenced by changes in rhizophore microbial community diversity and composition. SynCom induced the alteration of microbial community composition and diversity, which could partly explain the promotion of pepper growth in the current study. The changed rhizosphere community composition, diversity, and stability under SynCom conditions contributed to increased pepper growth and regulated the root morphology, which implied that the assembled rhizosphere microbial community play an important role in pepper growth. Specifically, the key native microbes, such as *Scedosporium*, *Sordariomycetes*, *Pseudarthrobacter*, *norank SBR1031*, and *norank A4b* may be pivotal in forming a distinctive rhizosphere microbial community by occupying the ecological niche, altering native microbes, and increasing microbial networks [47,48]. *Sordariomycetes* improve nutrient uptake via beneficial mycorrhizal associations, decompose organic matter to release essential nutrients, suppress diseases through secondary metabolites, and enhance stress tolerance by improving soil water retention. *Pseudarthrobacter* contributes by producing phytohormones that stimulate growth, solubilizing minerals to increase nutrient availability, fostering beneficial microbial interactions, and suppressing soil-borne pathogens. Together, these taxa create a synergistic environment in the rhizosphere, promoting nutrient cycling, disease resistance, and overall soil health, which is vital for sustainable agricultural productivity. Trivedi et al. reported that the application of SynCom inoculation can attract and enrich native beneficial microorganisms with growth-promoting functions and legacy effects, which could modulate the rhizosphere microbiome structure, fostering a potent microbial community with mutual promotional effects that collectively support plant growth and resistance [25]. The inoculation of *Proteobacteria*, *Actinobacteria*, *Firmicutes*, *Chloroflexia*, and *Zygomycota* increased the fungal community composition and diversity, and significantly enriched the proportion of saprophytic trophic fungi, with *Aspergillus* OTU3 identified as a key species in community assembly to promote plant growth by exhibiting a “synergistic effect” during the establishment of healthy soil [49,50]. It has been found that the combination of microbial communities can stimulate similar indigenous beneficial microbial communities and foster biofilm formation, secreting antioxidant proteins, glycosyl hydrolase family enzymes, β-glucosidase, expansion-like proteins, chitinase, and secondary metabolism proteins, and enhancing rhizosphere colonization [51]. Accordingly, SynCom inoculation can lead to microbial communities as observed in the current study. In addition, SynCom inoculation may play multiple roles to stimulate pepper growth, including assembling rhizosphere microbial communities, accelerating root vigor, and increasing root tips, thereby benefitting nutrient absorption and physiological metabolism [52]. These findings collectively suggest that the rhizosphere microbial community plays a promising role in driving plant development [53]. In terms of the current study, SynCom inoculation selectively promoted rhizosphere microbial community assembly and enriched the key taxa including *Scedosporium*, *Sordariomycetes*, *Pseudarthrobacter*, *norankSBR1031*, and *norankA4b* for pepper growth, which provide mechanistic insights into how the rhizosphere microbiome can be harnessed to regulate the growth and diversity of native microbes in the plant rhizosphere, thereby effectively promoting plant growth. In the future, we will delve deeper into examining the direct and indirect effects of key species on chili pepper growth. Exploring how these species influence microbial communities and their functions warrants further research, as it can offer specific support for targeted microbial selection. Additionally, inoculating synthetic microbial communities (SynComs) into field conditions encounters challenges due to environmental variability, including variations in soil types, climate factors, and plant interactions. Soil composition, such as texture and nutrient availability, influences microbial viability, while native microbial communities may compete with, or aid introduced strains. Fluctuations in temperature and moisture influence microbial activity, underscoring the importance of timing for application.

## 5. Conclusions

We found that SynCom inoculation significantly enhanced pepper growth, root vigor and root morphology. The mechanisms underlying the SynCom-promoted pepper growth include changed microbial diversity and enriched key native microbial taxa (Figure 6). First, SynCom inoculation would increase the Chao index and Bray–Curtis dissimilarity of the fungal community, and decrease the Bray–Curtis dissimilarity of the bacterial community, which influenced the microbial composition by creating a more favorable rhizosphere microbial community for the enrichment of key native taxa. Second, SynCom inoculation would selectively accelerate the key taxa abundance of *Scedosporium* (OTU451), *Sordariomycetes* (OTU346, OTU439, OTU437, OTU381, OTU447, and OTU182), *Pseudarthrobacter* (OTU2309), *norankSBR1031* (OTU3060, OTU2704), and *norankA4b* (OTU3638, OTU3534, OTU2118), which contribute to pepper growth and root morphology regulation. Therefore, SynCom inoculation provides an effective approach to promote pepper growth and regulate root morphology by modulating the diversity of rhizosphere microbial communities and increasing the abundance of key taxa like *Sordariomycetes* and *Pseudarthrobacter* at the seeding phase, thereby benefiting pepper health and productivity after transplanting.

## Figures and Tables

**Figure 1 microorganisms-13-00148-f001:**
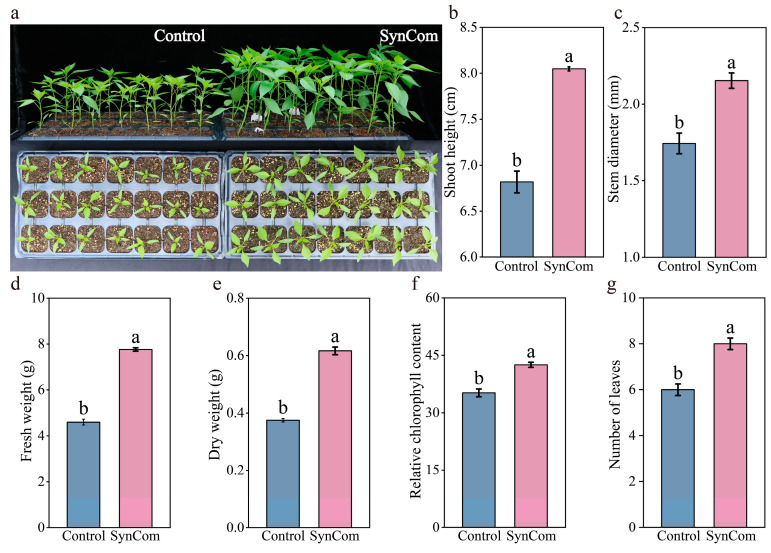
Effect of SynCom inoculation on pepper seeding growth. (**a**) Pepper seeding growth on the seeding substrate under SynCom inoculation after 45 days cultivation. (**b**) Effect of SynCom inoculation on shoot height of pepper plants. (**c**) Effect of SynCom inoculation on stem diameter of pepper plants. (**d**) Effect of SynCom inoculation on fresh weight of pepper plants. (**e**) Effect of SynCom inoculation on dry weight of pepper plants. (**f**) Effect of SynCom inoculation on chlorophyll content of pepper plants. (**g**) Effect of SynCom inoculation on the number of leaves. SynCom represents synthetic microbial community inoculation. Different letters indicate significant differences at *p* < 0.05 between SynCom inoculation and the control treatment.

**Figure 2 microorganisms-13-00148-f002:**
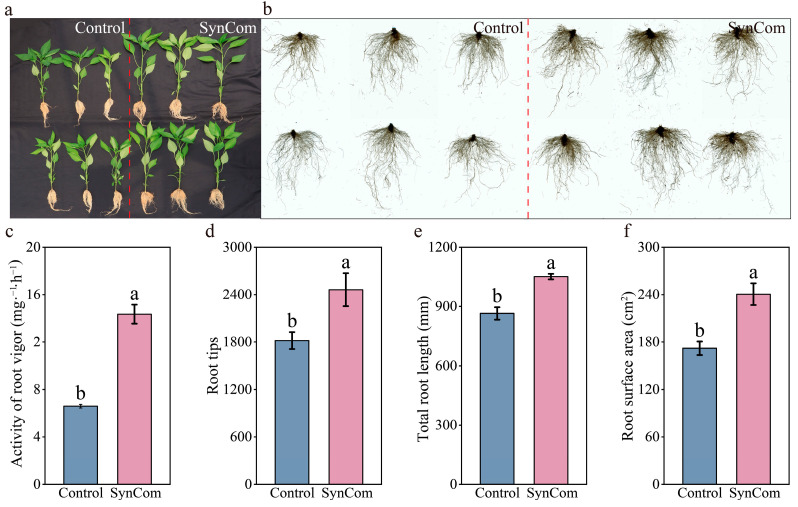
Effect of SynCom inoculation on pepper root morphology. (**a**) Effect of SynCom inoculation on root growth and development in the seedling substrate. (**b**) Effect of SynCom inoculation on root growth and development using a root scanner. (**c**) Effect of SynCom inoculation on the activity of root vigor in pepper plants. (**d**) Effect of SynCom inoculation on root tips in pepper plants. (**e**) Effect of SynCom inoculation on the total root length of pepper plants. (**f**) Effect of SynCom inoculation on the root surface area of pepper plants. SynCom represents synthetic microbial community inoculation. Different letters indicate significant differences at *p* < 0.05 between SynCom inoculation and the control treatment.

**Figure 3 microorganisms-13-00148-f003:**
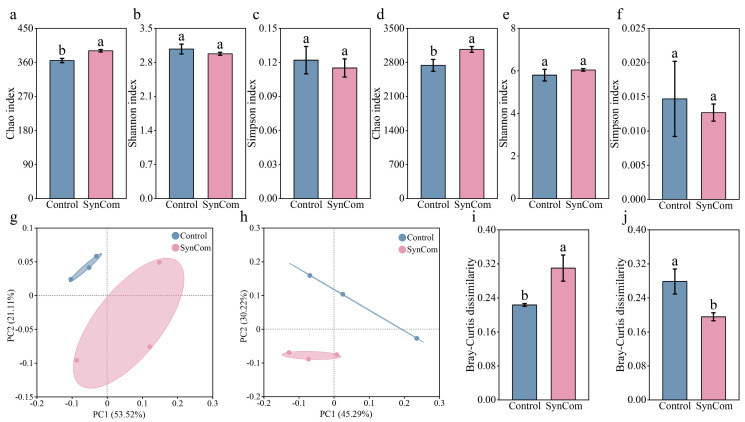
Effect of SynCom inoculation on the diversity of the rhizosphere microbial community. (**a**) Fungal Chao index for the two treatments. (**b**) Fungal Shannon index for the two treatments. (**c**) Fungal Simpson index for the two treatments. (**d**) Bacterial Chao index for the two treatments. (**e**) Bacterial Shannon index for the two treatments. (**f**) Bacterial Simpson index for the two treatments. (**g**) Beta diversity of the rhizosphere fungal community using PCoA principal component analysis. (**h**) Beta diversity of the rhizosphere bacterial community using PCoA principal component analysis. (**i**) Bray–Curtis dissimilarity of the rhizosphere fungal community. (**j**) Bray–Curtis dissimilarity of the rhizosphere bacterial community. SynCom represents synthetic microbial community inoculation. Different letters indicate significant differences at *p* < 0.05 between SynCom inoculation and the control treatment.

**Figure 4 microorganisms-13-00148-f004:**
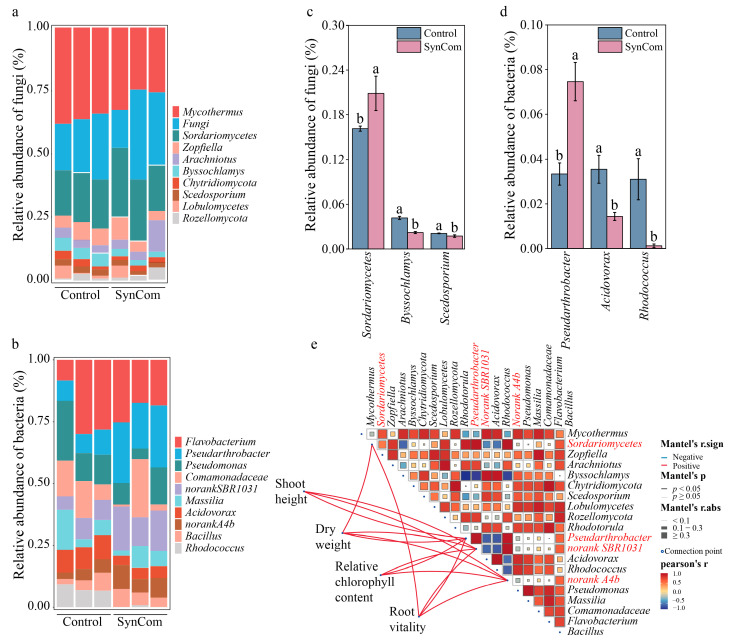
Effect of SynCom inoculation on the composition of the diversity of the rhizosphere microbial community. (**a**) Effect of SynCom on the composition of the fungal community at the genera level. (**b**) Effect of SynCom inoculation on the composition of the bacterial community at the genera level. (**c**) The relative abundance of significant differences in fungal at the genus level for the treatments. (**d**) The relative abundance of significant differences in bacterial genus for the treatments. (**e**) Correlation analysis between significant differences in species of microbial community composition and the growth and development of pepper for the two treatments. SynCom represents synthetic microbial community inoculation. Different letters indicate significant differences at *p* < 0.05 between SynCom inoculation and the control treatment.

**Figure 5 microorganisms-13-00148-f005:**
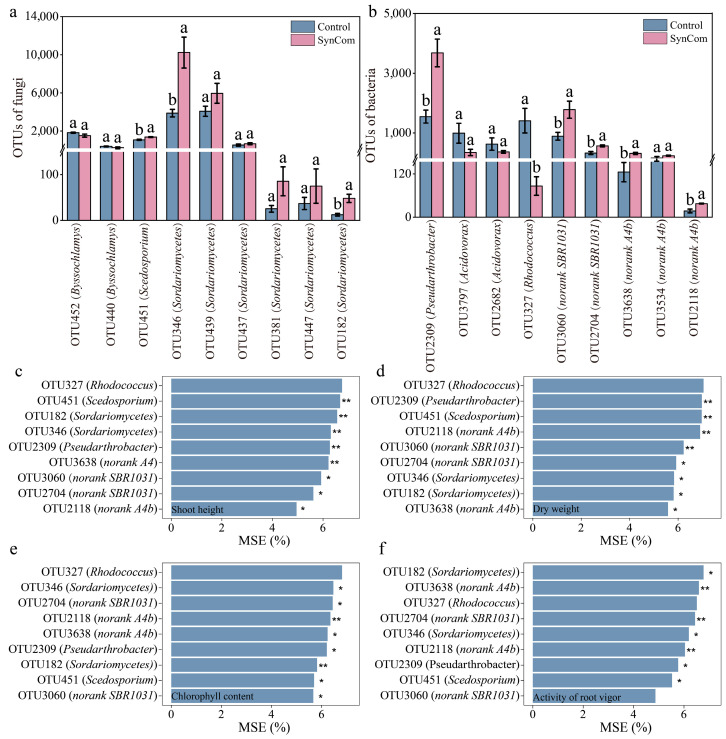
The contribution of microbial community characteristics to the growth and development of pepper seeding. (**a**) Relative abundance of fungal significant differences in OTUs for the two treatments. (**b**) Relative abundance of bacterial significant differences in OTUs for the two treatments. (**c**) Relative importance of significant OTUs in the microbial community on shoot height of pepper. (**d**) Relative importance of significant OTUs in the microbial community on dry weight of pepper. (**e**) Relative importance of significant OTUs in the microbial community on chlorophyll content of pepper. (**f**) Relative importance of significant OTUs in the microbial community on activity of root vigor. SynCom represents synthetic microbial community inoculation. Different letters indicate significant differences at *p* < 0.05 between SynCom inoculation and the control treatment. “*” indicated significant importance at *p* < 0.05, and “**” indicated significant importance at *p* < 0.01, between the SynCom inoculation and control.

**Figure 6 microorganisms-13-00148-f006:**
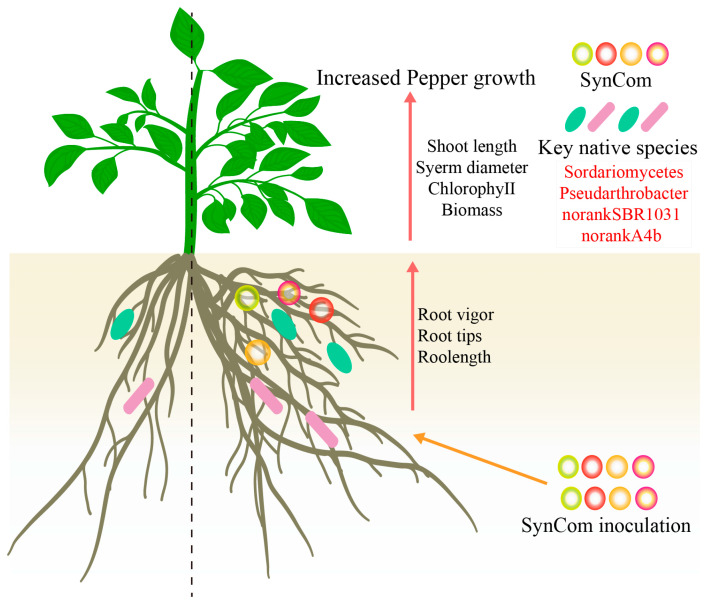
Schematic diagram illustrating the contribution of key microbial species to the promotion of pepper growth and root morphology. The upregulated parameters were labeled in red words, and the downregulated parameters were labeled in red words.

## Data Availability

The original contributions presented in this study are included in the article. Further inquiries can be directed to the corresponding authors.

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
