# Peer review of "Synthetic Microbial Communities Enhance Pepper Growth and Root Morphology by Regulating Rhizosphere Microbial Communities"

_microorganisms, 2025, doi:10.3390/microorganisms13010148_

Round 1

Reviewer 1 Report

Comments and Suggestions for Authors

Dear authors,

I studied with great interest the manuscript with the title „Synthetic Microbial Community Enhances Pepper Growth and Root Morphology by Regulating Rhizosphere Microbial Community”, SynCom application at the seedling stage enhances pepper growth by improving biomass, root architecture, and chlorophyll content (increases of 20.9%–117.42%). It modulates the rhizosphere microbiome, increasing fungal diversity and key taxa like Sordariomycetes and Pseudarthrobacter, which correlate with growth improvements. This highlights SynCom's potential for sustainable pepper cultivation.

The research presented in this manuscript is well-structured, clearly articulated, and successfully addresses its primary objectives. As a result, I support the publication of this manuscript in its current form without further revisions.

Congratulations to the authors.

Author Response

I studied with great interest the manuscript with the title „Synthetic Microbial Community Enhances Pepper Growth and Root Morphology by Regulating Rhizosphere Microbial Community”, SynCom application at the seedling stage enhances pepper growth by improving biomass, root architecture, and chlorophyll content (increases of 20.9%–117.42%). It modulates the rhizosphere microbiome, increasing fungal diversity and key taxa like Sordariomycetes and Pseudarthrobacter, which correlate with growth improvements. This highlights SynCom's potential for sustainable pepper cultivation.

The research presented in this manuscript is well-structured, clearly articulated, and successfully addresses its primary objectives. As a result, I support the publication of this manuscript in its current form without further revisions.

Congratulations to the authors.

Comments: no comments.

Response: Thanks for your positive approval for our work.

Reviewer 2 Report

Comments and Suggestions for Authors

The manuscript microorganisms-3417801, entitled “Synthetic microbial community enhances pepper growth and root morphology by regulating rhizosphere microbial community”, addresses an important paper investigating the effect of synthetic microbial communities (SynCom) on enhancing pepper (Capsicum annuum) growth and root morphology. However, in my opinion this paper must be revised in a major manner for reasons of forms and content.

Abstract

Could you elucidate the significance of the observed increases in growth parameters (e.g., shoot height and root length) in terms of agricultural productivity or sustainability?

Would it be helpful to briefly mention the experimental methods (e.g., microbial taxa involved) for context?

Introduction

Can you elaborate on the specific challenges faced in pepper cultivation that your study addresses, such as soil degradation or pest resistance?

Could you include recent advancements in SynCom applications for vegetable crops to better frame the novelty of your work?

The introduction mentions reduced SynCom functionality under complex field conditions. How does your study overcome these limitations?

Materials and Methods

Could you explain why the specific microbial strains (Bacillus subtilis, Trichoderma harzianum, etc.) were chosen for the SynCom? Were they based on prior studies or local soil compatibility?

Would additional details on replicates and statistical tests (e.g., corrections for multiple comparisons) improve reproducibility?

Did you include a control treatment with single microbial inoculants to compare SynCom’s synergistic effects?

Results

Your data show significant improvements in growth parameters. Can you provide a comparative context with traditional agricultural practices (e.g., chemical fertilizer use)?

The Bray-Curtis dissimilarity of microbial communities showed opposing trends for fungi and bacteria. Could you discuss why these trends occurred and their implications?

Were there any unexpected findings in the microbial taxa abundance or community composition that warrant further investigation?

Discussion

Could you explore how the enrichment of specific taxa (e.g., Sordariomycetes and Pseudarthrobacter) mechanistically contributes to improved growth?

Are there potential trade-offs, such as the effects of SynCom on non-target microbial populations or long-term soil health?

Can you discuss how the findings translate to field conditions, where environmental variability may affect SynCom efficacy?

Figures and Tables

Some figures compare multiple metrics (e.g., root morphology and microbial diversity). Could you consider integrating key insights from the figures into the main text for clarity?

Are the error bars in figures indicative of standard deviation or standard error? Clarifying this would aid interpretation.

Conclusion

The conclusion highlights SynCom's benefits but lacks actionable recommendations. Could you propose steps for integrating SynCom into large-scale agricultural practices?

Can you suggest future research directions, such as testing SynCom under different environmental or crop conditions?

Author Response

The manuscript microorganisms-3417801, entitled “Synthetic microbial community enhances pepper growth and root morphology by regulating rhizosphere microbial community”, addresses an important paper investigating the effect of synthetic microbial communities (SynCom) on enhancing pepper (Capsicum annuum) growth and root morphology. However, in my opinion this paper must be revised in a major manner for reasons of forms and content.

Thanks for the comments and suggestions. We have carefully revised the manuscript accordingly.

Abstract

Comments 1: Could you elucidate the significance of the observed increases in growth parameters (e.g., shoot height and root length) in terms of agricultural productivity or sustainability?

Response: Traditional agricultural practices, which heavily rely on chemical fertilizers and pesticides, have resulted in soil degradation, water pollution, and biodiversity loss, posing threats to long-term agricultural productivity and ecosystem health. Achieving agricultural sustainability necessitates reducing crops' dependence on chemical fertilizers, enhancing their capacity to thrive in marginal soil types, and bolstering their resilience against biotic and abiotic stressors. Microbial communities, particularly those associated with plants, play a crucial role in sustainable agriculture. The plant microbiome contributes to soil health, nutrient cycling, and plant resilience, making it a valuable asset in sustainable agricultural approaches. By harnessing the beneficial characteristics of these microbial communities, it is feasible to lessen reliance on chemical inputs and enhance crop performance across diverse environmental conditions. Synthetic microbial communities (SynComs) are a promising resource for developing ecology-based inoculants that enhance nutrient uptake, mitigate resistance stress against pathogens, and ensure sustainability by replicating the complexity and functionality of natural microbial communities, providing more stable and effective solutions for improving crop productivity and sustainability. Therefore, our findings suggest that SynCom inoculation can effectively enhance pepper growth and regulate root morphology by assembling rhizosphere microbial communities and increasing key taxa abundance like Sordariomycetes and Pseudarthrobacter, thereby benefiting nutrient acquisition, resistance enhancement, and pathogen resistance to ensure sustainability.

The concluding sentence of in the abstract was revised as “Our findings suggest that SynCom inoculation can effectively enhance pepper growth and regulating root morphology by regulating rhizosphere microbial communities and increasing key taxa abundance like Sordariomycetes and Pseudarthrobacter, thereby benefiting nutrient acquisition, resistance improvement, and pathogen resistance of crops to ensure sustainability.” (Lines 28-32).

Comments 2: Would it be helpful to briefly mention the experimental methods (e.g., microbial taxa involved) for context?

Response: Thanks for the great suggestion. We have provided detailed supplementation of the experimental methods in the Materials and Methods section. Due to the abstract words limit requirement stated in the "Information for Authors," the objectives and methods of the present study was briefly described.

Introduction

Comments 3: Can you elaborate on the specific challenges faced in pepper cultivation that your study addresses, such as soil degradation or pest resistance?

Response: Pepper cultivation faces several significant challenges that can impact yield and quality, including soil degradation, pest and disease management, water management, climate variability, market pressures, and sustainability practices. Peppers rank as the world's third-largest vegetable crop. In recent years, the global rise in temperature has led to a hot and humid climate, causing a surge in pepper diseases and pests such as anthracnose, spider mites, viral diseases, and blight. This has resulted in widespread seedling death and reduced yields, impacting the quality of peppers. With the development of intensive agriculture, improper use of fertilizers, pesticides, and plastic mulch has led to decreased pepper yield and quality, reduced soil fertility, disruption of the microecological structure, soil compaction, acidification, and outbreaks of diseases and pests. Additionally, due to the underdeveloped root system of peppers and their relatively poor nutrient absorption capacity, sustainable pepper production is under serious threat.

As suggested, a few sentences were added to show the specific challenges faced in pepper cultivation accordingly. (Lines 40-44).

Comments 4: Could you include recent advancements in SynCom applications for vegetable crops to better frame the novelty of your work?

Response: SynComs are artificially created microbial consortia composed of two or more distinct microorganisms with known taxonomic status and functional characteristics, mixed in specific proportions under defined conditions to obtain an efficient, powerful, controllable, easy to preserve, and easy to apply organic community. Compared to single microorganisms, they exhibit functional redundancy and are increasingly being applied in research on improving soil fertility, soil pollution bioremediation, suppression of soil-borne pathogens, and enhancement of soil resilience. Currently, SynComs have shown significant growth-promoting effects in crops such as tomatoes, cucumbers, and watermelons, primarily by increasing soil nitrogen content to enhance plant growth (Li et al., 2020), secreting plant growth regulators (such as auxins, gibberellins) to promote root development (Zapata et al., 2022), improving nutrient uptake efficiency, enhancing soil microbial community structure, and increasing soil biological activity (Islam et al., 2016). As suggested, a few sentences were added to show the SynCom applications on vegetable crops in revised manuscript (Lines 102-106).

Comments 5: The introduction mentions reduced SynCom functionality under complex field conditions. How does your study overcome these limitations?

Response: Many studies showed that the effectiveness of single microbial agent in promoting plant growth may be unstable due to their weak survival and colonization capacities in soil. Compared to single functional strains, the inoculation of SynCom into seedling substrate and soil has been found to be more efficient in promoting crop growth by broadening the types of metabolites and improving the synergistic effects of microbial community assembly. Therefore, we constructed a SynCom composed of Bacillus subtilis, Trichoderma harzianum, Trichoderma asperellum and Aspergillus sp.  And then the inoculation of a SynCom into pepper seedling substrates was conducted to assess its effectiveness in promoting pepper growth. The primary objectives were to address the following questions: (1) how efficient is the inoculation of the SynCom for promoting pepper growth, and (2) what are the characteristics of the rhizosphere microbial community responsible for enhancing pepper growth? We find that SynCom inoculation significantly enhanced pepper growth, root vigor and root morphology, which provides a effectively approach to promote pepper growth and regulate root morphology by modulating the diversity of rhizosphere microbial communities and increasing the abundance of key taxa like Sordariomycetes and Pseudarthrobacter at the seeding phase, thereby benefiting nutrient acquisition, resistance improvement, and pathogen resistance of crops to ensure sustainability.

Materials and Methods

Comments 6: Could you explain why the specific microbial strains (Bacillus subtilis, Trichoderma harzianum, etc.) were chosen for the SynCom? Were they based on prior studies or local soil compatibility?

Response: In our previous study, we collected healthy soil cultivated with long-term organic fertilizer and low-yield soil with chemical fertilizer from long term test field of Hunan Longping High tech Co., Ltd. Through high-throughput sequencing, we analyzed the core microorganisms that play a role in promotion pepper growth and disease resistance. Furthermore, through selective cultivation, we screened rhizosphere-promoting microorganisms that promote pepper growth and control pepper diseases such as phytophthora blight, fusarium wilt, and anthracnose. The results of pot experiment indicated that these rhizosphere-promoting microbes exhibited good disease resistance and yield-increasing effects on peppers.

Comments 7: Would additional details on replicates and statistical tests (e.g., corrections for multiple comparisons) improve reproducibility?

Response: The detailed description of replicates and statistical tests were rewritten as: “One hundred germinated seedlings with similar root and shoot sizes were placed in each tray containing 100 g of substrate for each treatment. Each treatment consists of three replicates of seedling trays. “Student’s T test (two-sided) was used to examine the difference in the variables between the two treatments. One-way analysis of variance (ANOVA) with the least significant difference (LSD) test was used to determine the difference of multiple comparisons.” (Lines 172-175; Lines 262-267).

Comments 8: Did you include a control treatment with single microbial inoculants to compare SynCom’s synergistic effects?

Response: Yes, we have detected the effect of single microbial inoculants to compare SynCom inoculation on pepper growth in green house with traditional agricultural practices. The effect of SynCom inoculation significant promoted pepper growth compared to single microbial inoculants. This section has not yet been published.

Many studies showed that the effectiveness of single microbial agent in promoting plant growth may be unstable due to their weak survival and colonization capacities in soil. Compared to single functional strains, the inoculation of SynCom into seedling substrate and soil has been found to be more efficient in promoting crop growth by broadening the types of metabolites and improving the synergistic effects of microbial community assembly. Therefore, this study constructed a SynCom composed of Bacillus subtilis, Trichoderma harzianum, Trichoderma asperellum and Aspergillus sp.  The primary objectives were to address the following questions: (1) how efficient is the inoculation of the SynCom for promoting pepper growth, and (2) what are the characteristics of the rhizosphere microbial community responsible for enhancing pepper growth?

Figure.1 The effect of different microbial inoculants on pepper growth

Results

Comments 9: Your data show significant improvements in growth parameters. Can you provide a comparative context with traditional agricultural practices (e.g., chemical fertilizer use)?

Response: Yes, we have detected the effect of SynCom inoculation on pepper growth in green house with traditional agricultural practices. The effect of SynCom inoculation significant promoted pepper growth. This section has not yet been published.

Figure. 2 The SynCom inoculation on pepper growth

Comments 10: The Bray-Curtis dissimilarity of microbial communities showed opposing trends for fungi and bacteria. Could you discuss why these trends occurred and their implications?

Response: As suggested, a few sentences were added as “The variations in Bray-Curtis dissimilarity trends between fungal and bacterial communities arise from their unique ecological niches and functions. Bacteria, with their quick responses to environmental changes due to shorter generation times, play diverse roles as decomposers and pathogens, while fungi, particularly in mycorrhizal associations, have longer lifespans and stable communities, sensitive to specific environ-mental conditions. These contrasting dynamics underscore the complexity of microbial interactions and responses to environmental factors, high-lighting the need for precise ecosystem management strategies to uphold ecosystem functioning.” (Lines 473-482).

Comments 11: Were there any unexpected findings in the microbial taxa abundance or community composition that warrant further investigation?

Response: We find that SynCom inoculation significantly enhanced pepper growth, root vigor and root morphology. The mechanisms underlying the SynCom-promoted pepper growth include changed microbial diversity and enriched key native microbial taxa (Fig. 6). First, SynCom inoculation would increase Chao index and Bray-Curtis dissimilarity of the fungal community, and decrease the Bray-Curtis dissimilarity of the bacterial community, which influenced the microbial composition by creating a more favorable rhizosphere microbial community for the enrichment of key native taxa. Second, SynCom inoculation would selectively accelerate the key taxa abundance of Scedosporium (OTU451), Sordariomycetes (OTU346, OTU439, OTU437, OTU381, OTU447, and OTU182), Pseudarthrobacter (OTU2309), norankSBR1031 (OTU3060, OTU2704) and norankA4b (OTU3638, OTU3534, OTU2118), which contributed to pepper growth and root morphology regulation. Therefore, SynCom inoculation provides a effectively approach to promote pepper growth and regulate root morphology by modulating the diversity of rhizosphere microbial communi-ties and increasing the abundance of key taxa like Sordariomycetes and Pseudarthrobacter at the seeding phase, thereby benefiting pepper health and productivity after transplanting. We believe that further research into the direct and indirect effects of key species on the growth of Capsicum annuum is warranted. Exploring how these species influence microbial communities and their functions merits in-depth investigation could provide precise support for targeted microbial selection.

Discussion

Comments 12: Could you explore how the enrichment of specific taxa (e.g., Sordariomycetes and Pseudarthrobacter) mechanistically contributes to improved growth?

Response: Some mechanisms underlying the effects of Sordariomycetes and Pseudarthrobacter on plant growth have already been provided as “Sordariomycetes improve nutrient uptake via beneficial mycorrhizal associations, decompose organic matter to release essential nutrients, suppress diseases through secondary metabolites, and enhance stress tolerance by improving soil water retention. Pseudarthrobacter contributes by producing phytohormones that stimulate growth, solubilizing minerals to increase nutrient availability, fostering beneficial microbial interactions, and suppressing soil-borne pathogens. Together, these taxa create a synergistic environment in the rhizosphere, promoting nutrient cycling, disease resistance, and overall soil health, which is vital for sustainable agricultural productivity.” (Lines 507-516).

Comments 13: Are there potential trade-offs, such as the effects of SynCom on non-target microbial populations or long-term soil health?

Response: Thanks for the great suggestions. The use of synthetic microbial communities (SynComs) in agriculture presents potential trade-offs, particularly regarding non-target microbial populations and long-term soil health. Introducing SynComs can disrupt native microbiomes, leading to decreased microbial diversity and competition for resources, which may diminish beneficial microbes essential for nutrient cycling and soil structure. Over-reliance on SynComs could result in homogenized microbial communities, negatively impacting soil resilience and nutrient balance. Additionally, the introduction of non-native microbes raises the risk of pathogen emergence. Therefore, while SynComs can enhance crop productivity, it is crucial to carefully consider their effects on existing ecosystems and conduct ongoing research and monitoring to ensure sustainable agricultural practices.

Comments 14: Can you discuss how the findings translate to field conditions, where environmental variability may affect SynCom efficacy?

Response: In the revised manuscript, more discussion wase provided as “In the future, we will delve deeper into examining the direct and indirect effects of key species on pepper growth. Exploring how these species influence microbial communities and their functions warrants further research, as it can offer specific support for targeted microbial selection. Additionally, inoculating synthetic microbial communities (SynComs) into field conditions encounters challenges due to environmental variability, including variations in soil types, climate factors, and plant interactions. Soil composition, such as texture and nutrient availability, influences microbial viability, while native microbial communities may compete with or aid introduced strains. Fluctuations in temperature and moisture influence microbial activity, underscoring the importance of timing for application”. (Lines 544-555).

Figures and Tables

Comments 15: Some figures compare multiple metrics (e.g., root morphology and microbial diversity). Could you consider integrating key insights from the figures into the main text for clarity?

Response: As suggested, a few sentences were integrated to the main text accordingly.

Comments 16: Are the error bars in figures indicative of standard deviation or standard error? Clarifying this would aid interpretation.

Response: Sorry for the confusion description. The error bars in figures represents standard error. A few sentences were added to in material and method to descript the error bars in revised manuscript.

Conclusion

Comments 17: The conclusion highlights SynCom's benefits but lacks actionable recommendations. Could you propose steps for integrating SynCom into large-scale agricultural practices?

Response: In the revised manuscript, more discussion wase provided as “In the future, we will delve deeper into examining the direct and indirect effects of key species on pepper growth. Exploring how these species influence microbial communities and their functions warrants further research, as it can offer specific support for targeted microbial selection. Additionally, inoculating synthetic microbial communities (SynComs) into field conditions encounters challenges due to environmental variability, including variations in soil types, climate factors, and plant interactions. Soil composition, such as texture and nutrient availability, influences microbial viability, while native microbial communities may compete with or aid introduced strains. Fluctuations in temperature and moisture influence microbial activity, underscoring the importance of timing for application”. (Lines 544-555).

Comments 18: Can you suggest future research directions, such as testing SynCom under different environmental or crop conditions?

Response: As suggested, a few sentences were added to suggest the future research directions of SynCom accordingly in revised manuscript. “Additionally, inoculating synthetic microbial communities (SynComs) into field conditions encounters challenges due to environmental varia-bility, including variations in soil types, climate factors, and plant inter-actions. Soil composition, such as texture and nutrient availability, influ-ences microbial viability, while native microbial communities may com-pete with or aid introduced strains. Fluctuations in temperature and moisture influence microbial activity, underscoring the importance of timing for application. (Lines 548-555).

Responses to Reviewer #2

The manuscript microorganisms-3417801, entitled “Synthetic microbial community enhances pepper growth and root morphology by regulating rhizosphere microbial community”, addresses an important paper investigating the effect of synthetic microbial communities (SynCom) on enhancing pepper (Capsicum annuum) growth and root morphology. However, in my opinion this paper must be revised in a major manner for reasons of forms and content.

Response: Thanks for the comments and suggestions. We have carefully revised the manuscript accordingly.

Abstract

Comments 1: Could you elucidate the significance of the observed increases in growth parameters (e.g., shoot height and root length) in terms of agricultural productivity or sustainability?

Response: Traditional agricultural practices, which heavily rely on chemical fertilizers and pesticides, have resulted in soil degradation, water pollution, and biodiversity loss, posing threats to long-term agricultural productivity and ecosystem health. Achieving agricultural sustainability necessitates reducing crops' dependence on chemical fertilizers, enhancing their capacity to thrive in marginal soil types, and bolstering their resilience against biotic and abiotic stressors. Microbial communities, particularly those associated with plants, play a crucial role in sustainable agriculture. The plant microbiome contributes to soil health, nutrient cycling, and plant resilience, making it a valuable asset in sustainable agricultural approaches. By harnessing the beneficial characteristics of these microbial communities, it is feasible to lessen reliance on chemical inputs and enhance crop performance across diverse environmental conditions. Synthetic microbial communities (SynComs) are a promising resource for developing ecology-based inoculants that enhance nutrient uptake, mitigate resistance stress against pathogens, and ensure sustainability by replicating the complexity and functionality of natural microbial communities, providing more stable and effective solutions for improving crop productivity and sustainability. Therefore, our findings suggest that SynCom inoculation can effectively enhance pepper growth and regulate root morphology by assembling rhizosphere microbial communities and increasing key taxa abundance like Sordariomycetes and Pseudarthrobacter, thereby benefiting nutrient acquisition, resistance enhancement, and pathogen resistance to ensure sustainability.

The concluding sentence of in the abstract was revised as “Our findings suggest that SynCom inoculation can effectively enhance pepper growth and regulating root morphology by regulating rhizosphere microbial communities and increasing key taxa abundance like Sordariomycetes and Pseudarthrobacter, thereby benefiting nutrient acquisition, resistance improvement, and pathogen resistance of crops to ensure sustainability.” (Lines 28-32).

Comments 2: Would it be helpful to briefly mention the experimental methods (e.g., microbial taxa involved) for context?

Response: Thanks for the great suggestion. We have provided detailed supplementation of the experimental methods in the Materials and Methods section. Due to the abstract words limit requirement stated in the "Information for Authors," the objectives and methods of the present study was briefly described.

Introduction

Comments 3: Can you elaborate on the specific challenges faced in pepper cultivation that your study addresses, such as soil degradation or pest resistance?

Response: Pepper cultivation faces several significant challenges that can impact yield and quality, including soil degradation, pest and disease management, water management, climate variability, market pressures, and sustainability practices. Peppers rank as the world's third-largest vegetable crop. In recent years, the global rise in temperature has led to a hot and humid climate, causing a surge in pepper diseases and pests such as anthracnose, spider mites, viral diseases, and blight. This has resulted in widespread seedling death and reduced yields, impacting the quality of peppers. With the development of intensive agriculture, improper use of fertilizers, pesticides, and plastic mulch has led to decreased pepper yield and quality, reduced soil fertility, disruption of the microecological structure, soil compaction, acidification, and outbreaks of diseases and pests. Additionally, due to the underdeveloped root system of peppers and their relatively poor nutrient absorption capacity, sustainable pepper production is under serious threat.

As suggested, a few sentences were added to show the specific challenges faced in pepper cultivation accordingly. (Lines 40-44).

Comments 4: Could you include recent advancements in SynCom applications for vegetable crops to better frame the novelty of your work?

Response: SynComs are artificially created microbial consortia composed of two or more distinct microorganisms with known taxonomic status and functional characteristics, mixed in specific proportions under defined conditions to obtain an efficient, powerful, controllable, easy to preserve, and easy to apply organic community. Compared to single microorganisms, they exhibit functional redundancy and are increasingly being applied in research on improving soil fertility, soil pollution bioremediation, suppression of soil-borne pathogens, and enhancement of soil resilience. Currently, SynComs have shown significant growth-promoting effects in crops such as tomatoes, cucumbers, and watermelons, primarily by increasing soil nitrogen content to enhance plant growth (Li et al., 2020), secreting plant growth regulators (such as auxins, gibberellins) to promote root development (Zapata et al., 2022), improving nutrient uptake efficiency, enhancing soil microbial community structure, and increasing soil biological activity (Islam et al., 2016). As suggested, a few sentences were added to show the SynCom applications on vegetable crops in revised manuscript (Lines 102-106).

Comments 5 The introduction mentions reduced SynCom functionality under complex field conditions. How does your study overcome these limitations?

Response: Many studies showed that the effectiveness of single microbial agent in promoting plant growth may be unstable due to their weak survival and colonization capacities in soil. Compared to single functional strains, the inoculation of SynCom into seedling substrate and soil has been found to be more efficient in promoting crop growth by broadening the types of metabolites and improving the synergistic effects of microbial community assembly. Therefore, we constructed a SynCom composed of Bacillus subtilis, Trichoderma harzianum, Trichoderma asperellum and Aspergillus sp.  And then the inoculation of a SynCom into pepper seedling substrates was conducted to assess its effectiveness in promoting pepper growth. The primary objectives were to address the following questions: (1) how efficient is the inoculation of the SynCom for promoting pepper growth, and (2) what are the characteristics of the rhizosphere microbial community responsible for enhancing pepper growth? We find that SynCom inoculation significantly enhanced pepper growth, root vigor and root morphology, which provides a effectively approach to promote pepper growth and regulate root morphology by modulating the diversity of rhizosphere microbial communities and increasing the abundance of key taxa like Sordariomycetes and Pseudarthrobacter at the seeding phase, thereby benefiting nutrient acquisition, resistance improvement, and pathogen resistance of crops to ensure sustainability.

Materials and Methods

Comments 6: Could you explain why the specific microbial strains (Bacillus subtilis, Trichoderma harzianum, etc.) were chosen for the SynCom? Were they based on prior studies or local soil compatibility?

Response: In our previous study, we collected healthy soil cultivated with long-term organic fertilizer and low-yield soil with chemical fertilizer from long term test field of Hunan Longping High tech Co., Ltd. Through high-throughput sequencing, we analyzed the core microorganisms that play a role in promotion pepper growth and disease resistance. Furthermore, through selective cultivation, we screened rhizosphere-promoting microorganisms that promote pepper growth and control pepper diseases such as phytophthora blight, fusarium wilt, and anthracnose. The results of pot experiment indicated that these rhizosphere-promoting microbes exhibited good disease resistance and yield-increasing effects on peppers.

Comments 7: Would additional details on replicates and statistical tests (e.g., corrections for multiple comparisons) improve reproducibility?

Response: The detailed description of replicates and statistical tests were rewritten as: “One hundred germinated seedlings with similar root and shoot sizes were placed in each tray containing 100 g of substrate for each treatment. Each treatment consists of three replicates of seedling trays. “Student’s T test (two-sided) was used to examine the difference in the variables between the two treatments. One-way analysis of variance (ANOVA) with the least significant difference (LSD) test was used to determine the difference of multiple comparisons.” (Lines 172-175; Lines 262-267).

Comments 8: Did you include a control treatment with single microbial inoculants to compare SynCom’s synergistic effects?

Response: Yes, we have detected the effect of single microbial inoculants to compare SynCom inoculation on pepper growth in green house with traditional agricultural practices. The effect of SynCom inoculation significant promoted pepper growth compared to single microbial inoculants. This section has not yet been published.

Many studies showed that the effectiveness of single microbial agent in promoting plant growth may be unstable due to their weak survival and colonization capacities in soil. Compared to single functional strains, the inoculation of SynCom into seedling substrate and soil has been found to be more efficient in promoting crop growth by broadening the types of metabolites and improving the synergistic effects of microbial community assembly. Therefore, this study constructed a SynCom composed of Bacillus subtilis, Trichoderma harzianum, Trichoderma asperellum and Aspergillus sp.  The primary objectives were to address the following questions: (1) how efficient is the inoculation of the SynCom for promoting pepper growth, and (2) what are the characteristics of the rhizosphere microbial community responsible for enhancing pepper growth?

Figure.1 The effect of different microbial inoculants on pepper growth

Results

Comments 9: Your data show significant improvements in growth parameters. Can you provide a comparative context with traditional agricultural practices (e.g., chemical fertilizer use)?

Response: Yes, we have detected the effect of SynCom inoculation on pepper growth in green house with traditional agricultural practices. The effect of SynCom inoculation significant promoted pepper growth. This section has not yet been published.

Figure. 2 The SynCom inoculation on pepper growth

Comments 10: The Bray-Curtis dissimilarity of microbial communities showed opposing trends for fungi and bacteria. Could you discuss why these trends occurred and their implications?

Response: As suggested, a few sentences were added as “The variations in Bray-Curtis dissimilarity trends between fungal and bacterial communities arise from their unique ecological niches and functions. Bacteria, with their quick responses to environmental changes due to shorter generation times, play diverse roles as decomposers and pathogens, while fungi, particularly in mycorrhizal associations, have longer lifespans and stable communities, sensitive to specific environ-mental conditions. These contrasting dynamics underscore the complexity of microbial interactions and responses to environmental factors, high-lighting the need for precise ecosystem management strategies to uphold ecosystem functioning.” (Lines 473-482).

Comments 11: Were there any unexpected findings in the microbial taxa abundance or community composition that warrant further investigation?

Response: We find that SynCom inoculation significantly enhanced pepper growth, root vigor and root morphology. The mechanisms underlying the SynCom-promoted pepper growth include changed microbial diversity and enriched key native microbial taxa (Fig. 6). First, SynCom inoculation would increase Chao index and Bray-Curtis dissimilarity of the fungal community, and decrease the Bray-Curtis dissimilarity of the bacterial community, which influenced the microbial composition by creating a more favorable rhizosphere microbial community for the enrichment of key native taxa. Second, SynCom inoculation would selectively accelerate the key taxa abundance of Scedosporium (OTU451), Sordariomycetes (OTU346, OTU439, OTU437, OTU381, OTU447, and OTU182), Pseudarthrobacter (OTU2309), norankSBR1031 (OTU3060, OTU2704) and norankA4b (OTU3638, OTU3534, OTU2118), which contributed to pepper growth and root morphology regulation. Therefore, SynCom inoculation provides a effectively approach to promote pepper growth and regulate root morphology by modulating the diversity of rhizosphere microbial communi-ties and increasing the abundance of key taxa like Sordariomycetes and Pseudarthrobacter at the seeding phase, thereby benefiting pepper health and productivity after transplanting. We believe that further research into the direct and indirect effects of key species on the growth of Capsicum annuum is warranted. Exploring how these species influence microbial communities and their functions merits in-depth investigation could provide precise support for targeted microbial selection.

Discussion

Comments 12: Could you explore how the enrichment of specific taxa (e.g., Sordariomycetes and Pseudarthrobacter) mechanistically contributes to improved growth?

Response: Some mechanisms underlying the effects of Sordariomycetes and Pseudarthrobacter on plant growth have already been provided as “Sordariomycetes improve nutrient uptake via beneficial mycorrhizal associations, decompose organic matter to release essential nutrients, suppress diseases through secondary metabolites, and enhance stress tolerance by improving soil water retention. Pseudarthrobacter contributes by producing phytohormones that stimulate growth, solubilizing minerals to increase nutrient availability, fostering beneficial microbial interactions, and suppressing soil-borne pathogens. Together, these taxa create a synergistic environment in the rhizosphere, promoting nutrient cycling, disease resistance, and overall soil health, which is vital for sustainable agricultural productivity.” (Lines 507-516).

Comments 13: Are there potential trade-offs, such as the effects of SynCom on non-target microbial populations or long-term soil health?

Response: Thanks for the great suggestions. The use of synthetic microbial communities (SynComs) in agriculture presents potential trade-offs, particularly regarding non-target microbial populations and long-term soil health. Introducing SynComs can disrupt native microbiomes, leading to decreased microbial diversity and competition for resources, which may diminish beneficial microbes essential for nutrient cycling and soil structure. Over-reliance on SynComs could result in homogenized microbial communities, negatively impacting soil resilience and nutrient balance. Additionally, the introduction of non-native microbes raises the risk of pathogen emergence. Therefore, while SynComs can enhance crop productivity, it is crucial to carefully consider their effects on existing ecosystems and conduct ongoing research and monitoring to ensure sustainable agricultural practices.

Comments 14: Can you discuss how the findings translate to field conditions, where environmental variability may affect SynCom efficacy?

Response: In the revised manuscript, more discussion wase provided as “In the future, we will delve deeper into examining the direct and indirect effects of key species on pepper growth. Exploring how these species influence microbial communities and their functions warrants further research, as it can offer specific support for targeted microbial selection. Additionally, inoculating synthetic microbial communities (SynComs) into field conditions encounters challenges due to environmental variability, including variations in soil types, climate factors, and plant interactions. Soil composition, such as texture and nutrient availability, influences microbial viability, while native microbial communities may compete with or aid introduced strains. Fluctuations in temperature and moisture influence microbial activity, underscoring the importance of timing for application”. (Lines 544-555).

Figures and Tables

Comments 15: Some figures compare multiple metrics (e.g., root morphology and microbial diversity). Could you consider integrating key insights from the figures into the main text for clarity?

Response: As suggested, a few sentences were integrated to the main text accordingly.

Comments 16: Are the error bars in figures indicative of standard deviation or standard error? Clarifying this would aid interpretation.

Response: Sorry for the confusion description. The error bars in figures represents standard error. A few sentences were added to in material and method to descript the error bars in revised manuscript.

Conclusion

Comments 17: The conclusion highlights SynCom's benefits but lacks actionable recommendations. Could you propose steps for integrating SynCom into large-scale agricultural practices?

Response: In the revised manuscript, more discussion wase provided as “In the future, we will delve deeper into examining the direct and indirect effects of key species on pepper growth. Exploring how these species influence microbial communities and their functions warrants further research, as it can offer specific support for targeted microbial selection. Additionally, inoculating synthetic microbial communities (SynComs) into field conditions encounters challenges due to environmental variability, including variations in soil types, climate factors, and plant interactions. Soil composition, such as texture and nutrient availability, influences microbial viability, while native microbial communities may compete with or aid introduced strains. Fluctuations in temperature and moisture influence microbial activity, underscoring the importance of timing for application”. (Lines 544-555).

Comments 18: Can you suggest future research directions, such as testing SynCom under different environmental or crop conditions?

Response: As suggested, a few sentences were added to suggest the future research directions of SynCom accordingly in revised manuscript. “Additionally, inoculating synthetic microbial communities (SynComs) into field conditions encounters challenges due to environmental varia-bility, including variations in soil types, climate factors, and plant inter-actions. Soil composition, such as texture and nutrient availability, influ-ences microbial viability, while native microbial communities may compete with or aid introduced strains. Fluctuations in temperature and moisture influence microbial activity, underscoring the importance of timing for application. (Lines 548-555).

Round 2

Reviewer 2 Report

Comments and Suggestions for Authors

I am writing to follow up on my review of manuscript microorganisms-3417801, entitled: Synthetic microbial community enhances pepper growth and root morphology by regulating rhizosphere microbial community. I'm pleased to inform you that the authors have addressed all my comments thoughtfully and respectfully.

Their revisions demonstrate a clear understanding of the points I raised, and the changes they have made significantly improve the quality of the manuscript. I believe the paper is now much stronger and ready for further consideration.